# Sizing ice hydrometeor populations using dual-wavelength radar ratio

Sergey Y. Matrosov[1,2], Alexei Korolev[3], Mengistu Wolde[4], Cuong Nguyen[4]

[1]Cooperative Institute for Research in Environmental Sciences, University of Colorado, Boulder, CO, 803039, USA
[2]National Atmospheric and Oceanic Administration, Physical Sciences Laboratory, Boulder, CO, 80305, USA
[3]Environment and Climate Change Canada, Toronto, ON, M3H5T4, Canada
[4]Flight Research Laboratory, National Research Council Canada, Ottawa, K1A0R6, Canada

*Correspondence to*: Sergey Y. Matrosov (sergey.matrosov@noaa.gov)

**Abstract.**    Dual-wavelength (3.2 and 0.32 cm, i.e., X- and W- radar bands) radar ratio (DWR) measurements in ice clouds and precipitation using the Canada's National Research Council Institute for Aerospace Research airborne radar are compared to closely collocated particle microphysical *in situ* sampling data in order to develop relations between DWR and characteristic hydrometeor size. This study uses the radar and *in situ* data sets collected during the In-Cloud ICing and Large-drop Experiment (ICICLE) campaign in mid-latitude frontal clouds. Since atmospheric particle scattering at X-band is predominantly in the Rayleigh regime and the W-band frequency is the highest frequency usually used for hydrometeor remote sensing, the X-W-band combination provides relatively strong dual-wavelength reflectivity difference. This study considers radar and *in situ* measurements conducted in relatively homogeneous cloud and precipitation conditions. Measurements show that under these conditions, the difference between the X-band radar reflectivities observed with vertical and horizontal pointing of the radar beam are generally small and often negligible. However, W-band reflectivities at vertical beam pointing are, on average, larger than those for horizontal beam pointing by about 4 dB, which is a non-Rayleigh scattering effect from preferentially oriented non-spherical particles. A horizontal radar beam DWR – mean volume particle size, $D_v$, relation provides robust estimates of this characteristic size for populations of particles with different habits. Uncertainties of $D_v$ retrievals using DWR are around 0.6 mm when $D_v$ is greater than approximately 1 mm. Size estimates using vertical radar beam DWRs have larger uncertainties due to smaller dual-wavelength signals and stronger influences of hydrometeor habits and orientations at this geometry of beam pointing. Mean relations among different characteristic sizes, which describe the entire particle size distribution (PSD), such as $D_v$, and other sizes used in various applications (e.g., the mean, effective, and median sizes) are derived, so the results of this study can be used for estimating different PSD characteristic sizes.

## 1 Introduction

Multi-wavelength meteorological radar systems are useful tools for studies of clouds and precipitation. These systems are deployed at different locations such as ground-based facilities of the U.S. Department of Energy's Atmospheric Radiation

Measurement (ARM) sites (e.g., Kollias et al. 2014) as well as on airborne platforms (e.g., Heymsfield et al. 2016; Nguyen et al. 2021). The availability of simultaneous and approximately collocated radar measurements at different frequencies stimulated interest in the development of remote sensing approaches for hydrometeor property retrievals using multi-wavelength radar data.

Compared to single wavelength measurements, multi-wavelength radar measurements of clouds and precipitation provide additional information on hydrometeor properties when deviations from the Rayleigh-scattering regime (i.e., the scattering regime when scatterer sizes are much smaller than the radar wavelength) are different at different wavelengths. In the earlier studies of the multi-wavelength approach, it was shown (e.g., Matrosov 1993; Hogan et al. 2000; Liao et al. 2005) that measurements at two wavelengths can be used to infer information on characteristic sizes of the ice hydrometeor populations. This information is important for many applications since the cloud feedback in models is strongly affected by the hydrometeor sizes (Tan and Storelvmo 2019). Relatively recently, triple-wavelength radar approaches were developed for retrievals of hydrometeor properties and information on the ice cloud and precipitation processes, such as particle riming and aggregation (e.g., Kneifel et al. 2011; Tyynelä and Chandrasekar 2014; Leinonen et al. 2018; Mason et al. 2019; Tridon et al. 2019; Mroz et al. 2021). Collocated triple-wavelength measurement systems, are, however, not that widely available compared to dual-wavelength ones.

Reliable *in situ* information about the hydrometeor properties estimated from radar measurements is necessary for the assessment of the accuracy of remote sensing retrievals. To mitigate the effect of natural spatial inhomogeneity of cloud miscrostructure, it is important that the radar and *in situ* microphysical measurements are temporally and spatially collocated to the best possible extent. It can be achieved when both radar systems and *in situ* instruments are deployed on the same platform. One example of such platforms is the National Research Council of Canada (NRC) twin-engine research aircraft Convair-580 aircraft, which is equipped with collocated W-(94.05 GHz) and X-(9.41 GHz) band radars (Wolde and Pazmany 2005; Wolde et al. 2016; Nguyen et al. 2021) and a state-of-the-art *in situ* microphysical instrumentation.

The data set used in this study was collected during the Federal Aviation Administration (FAA) In-Cloud ICing and Large-drop Experiment (ICICLE) campaign conducted in January-March 2019 (Bernstein et al. 2021). During this campaign, the NRC, in collaboration with FAA and Environment and Climate Change Canada (ECCC), collected *in situ* and remote sensing measurements from the NRC Convair-580 aircraft in icing conditions. The flight operations were performed out of Rockford, Illinois. Although these studies primarily targeted supercooled liquid clouds, the data set obtained during the ICICLE campaign includes a variety of ice clouds and precipitation.

The main objective of this study was to develop and assess statistical relations between the dual-wavelength radar ratio and characteristic sizes of ice hydrometeor populations. Since recent studies (e.g., Matrosov et al. 2019) indicated the dependence of dual-wavelength radar signatures on the pointing of the radar beam, another important task pursued by this study was evaluating influences of viewing geometry on radar measurements and particle size retrievals. Other objectives included assessment of the ability of single frequency radar measurements to infer characteristic particle size and establishing statistical relations between various definitions of characteristic particle sizes used for describing hydrometeor populations.

## 2 Measurements

### 2.1 Instrumentation

During the ICICLE campaign, the airborne measurements of the radar reflectivity factor (hereafter, just reflectivity) measurements aboard the Convair-580 aircraft were conducted using the NRC Airborne W- and X-bands (NAWX) radar system (Wolde and Pazmany 2005). The NAWX antenna configuration allows for simultaneous measurements of radar returns from three directions: (a) sideward (i.e., horizontal), (b) upward, and (c) downward directions. Horizontal polarization measurements of X-band (i.e., $Z_{eX}$) and W-band (i.e., $Z_{eW}$) equivalent reflectivity were further used in this study for all radar beam pointing directions. The dual-wavelength ratio - DWR (also sometimes referred to as the dual frequency ratio – DFR) is expressed in the logarithmic units as

$$DWR \text{ (dB)} = Z_{eX}(dBZ) - Z_{eW}(dBZ). \quad\quad\quad (1)$$

Except in heavier precipitation, hydrometeor scattering at X-band frequency is mostly within the Rayleigh scattering regime (e.g., Matrosov et al. 2014). Some deviations from this regime occur for larger ice particles, whose sizes are greater than approximately 5 mm. With this lower frequency, the use of the higher W-band frequency provides the largest wavelength separation compared to other commonly used cloud radar frequencies [e.g., Ka-band (~35 GHZ), Ku-band (~14 GHz)]. This results in stronger and more pronounced DWR signals compared to other cloud radar frequency pairs.

The airborne *in situ* microphysical measurements were performed with an advanced suite of microphysical sensors. The Droplet Measurement Technologies (DMT) Cloud Droplet Probe (CDP, Lance et al. 2010) and the Stratton Park Engineering Company (SPEC) Fast Cloud Droplet Probe (FCDP) (Lawson et al. 2017) were used for measurements of particles in the size range from 2 to 50 $\mu$m. The two-dimensional cloud optical array probe (OAP-2DC) (Knollenberg 1981) was used to measure particles in the 50 $\mu$m – 1.6 mm range. The SPEC 2D imaging-stereo (2D-S) probe covered measurements of cloud particles in the nominal size range from 10 to 1280 $\mu$m (Lawson et al. 2006). The SPEC High Volume Precipitation Spectrometer (HVPS) was employed for measurements of particles in the nominal size range from 200 $\mu$m to 1.92 cm (Lawson et al. 2017). The processing software applied retrieval algorithms of partially viewed particle images (Heymsfield and Parrish 1979; Korolev and Sussman 2000), which allowed the enhancement of particle statistics and extended the maximum size of the composite particle size distribution (PSD) up ~ 3.8 cm. All particle probes were equipped with anti-shattering tips to mitigate the effect of ice shattering on the measurements of ice particle concentrations. The residual shattering artefacts were identified and filtered out with the help of the inter-arrival time algorithm (Field et al. 2006; Korolev and Field 2015). Calculations of composite PSDs for ice cloud segments employed 2D-S and HVPS measurements in the following size subranges (the midpoints are shown): 40 $\mu$m to 670 $\mu$m (at 10 $\mu$m resolution) and 750 $\mu$m to 3.84 cm (at 150 $\mu$m resolution), respectively. Liquid versus ice hydrometeor type identification was performed based on the analysis of measurements of a combination of the Rosemount Icing Detector (RID, Mazin et al. 2001), the particle scattering probes (CDP, FCDP), the Cloud Particle Imager (CPI, Lawson et al. 2001) and the 2D-S probe. The analysis of the RID and scattering probes measurements was primarily used to identify the presence of small liquid cloud droplets. The CPI and 2D-S imagery was used to identify drizzle or rain.

## 2.2 Methodology and data sets

Of the main interest to this study were ice hydrometeors observed at temperatures below freezing in relatively spatially homogeneous cloud conditions, so the relations between DWR and reflectivities observed at a distance of several hundred meters, where radar measurements are reliable, and microphysical data aboard the aircraft can be robustly assessed. The selected data excluded in-cloud segments in the proximity of cloud boundaries. This ensured availability of meaningful radar measurements at both frequencies for all radar beam pointing directions up to a 520 m range gate. Such conditions were observed for extended periods (>1 h) during ICICLE flights 5, 9, 15, 16, 20, 29, and they were further used in this study. The vertical beam pointing reflectivity is defined as an average between reflectivities measured at upward and downward directions. The averaging was performed in the linear units of $mm^6 m^{-3}$. The radar and microphysical data were averaged in 30-second intervals. Deviations of upward and downward radar beam pointing from the true vertical direction generally do not exceed few degrees and they were neglected. The radar and microphysical *in situ* data are available from Nguyen and Wolde (2020a, 2020b) and Korolev and Heckman (2020a, 2020b). The microphysical data were processed using the ECCC D2G software.

In order to minimize the influence of supercooled water drops on reflectivity measurements and to avoid the contamination by the melting layer particles, the time periods when the temperature at the aircraft level was higher than -2°C were excluded. Data collected during mixed-phase periods when estimates of liquid drop mean volume diameter (MVD) were greater than 10% of ice particle MVD values and liquid water content values were larger than 0.05 g $m^{-3}$ were also excluded. These exclusions ensure that the liquid phase contribution (if any) to the total reflectivity is generally negligible and the radar echoes are dominated by ice hydrometeors. Furthermore, since this study considered radar measurements in an ice cloud environment at close ranges, the radar signal attenuation was neglected.

The 250 m range is the nearest to the aircraft, where NAWX radar measurements can be considered reliable (Nguyen et al. 2021). Therefore, the radar and *in situ* data were spatially separated by 250 m. Since objectives of this study included establishing relations between the radar measurements and cloud microphysical parameters, it is important that clouds are relatively homogeneous at scales of several hundred meters. In order to evaluate homogeneity of clouds for the data set considered in this study, Fig. 1 shows the data scatter between reflectivities observed with the horizontal radar beam pointing at a 250 m and 520 m distances from the aircraft. As seen from this figure, correlations between 250 m and 520 m reflectivities are very high ($\geq 0.995$) at both frequencies. The root-mean square deviation (RMSD) and bias values between 250 and 520 m reflectivities are generally small and on the order of radar measurement uncertainties. Such spatial homogeneity of radar reflectivity satisfies the objectives of this study.

Characteristic sizes of PSDs are often defined as a ratio of different PSD moments. The PSD N-th moment is defined as

$$M_N = \sum_i D_i^N n(D_i) \Delta D_i, \qquad (2)$$

where $D_i$, $\Delta D_i$, and $n(D_i)$ are the bin centre, bin width, and particle concentration per a particle size unit in the i-th bin of the composite ice PSD measured by the microphysical probes, and the summation is performed over all bin sizes. Individual particle sizes, $D_i$, are given in terms of the major dimensions of 2D images of the particles. The X-W-band DWR – characteristic sizes relations in this study are sought in terms of the mean volume particle size (also sometimes referred to as a volume moment mean size):

$$D_v = M_4/M_3. \tag{3}$$

This size is retrieved in a number of polarimetric radar remote sensing approaches (e.g., Ryzhkov et al. 2018). Mean statistical relations among various characteristic sizes used in different direct measurements and remote sensing approaches are discussed in section 5.

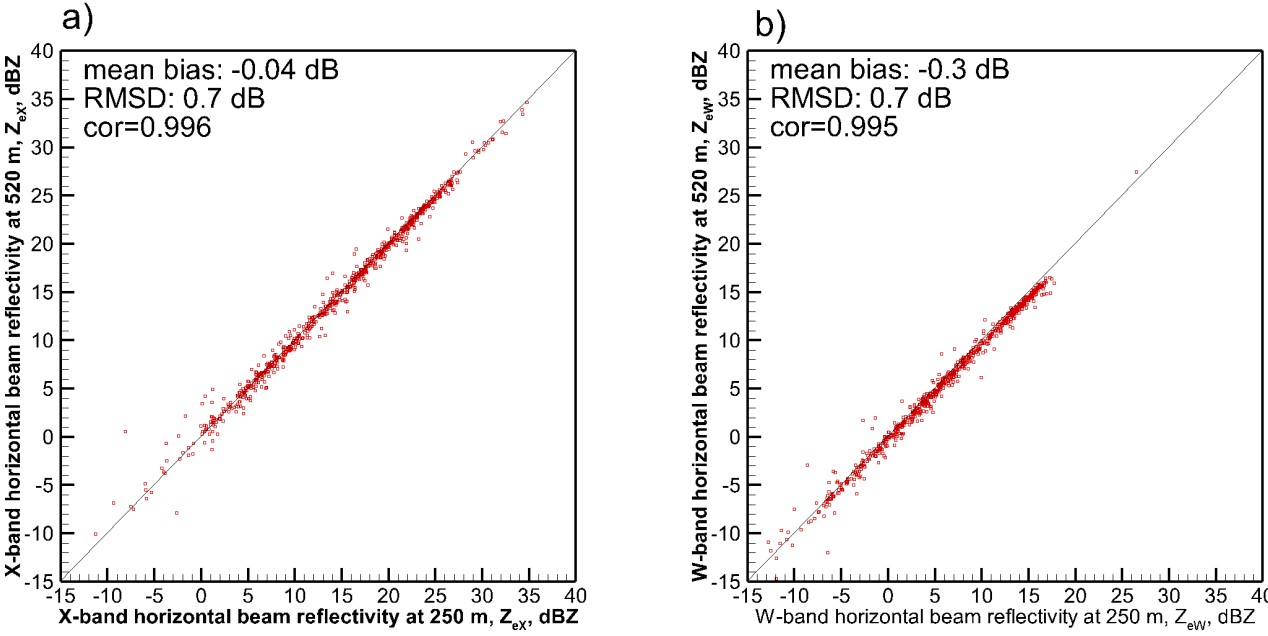

**Figure 1: Scatter plots of observed reflectivities at 250 and 520 m distances for (a) X-band and (b) W-band horizontal beam data**

## 2 A case study illustration

Figure 2a shows time series of X- and W-band reflectivity measurements for horizontal and vertical beam pointing at a 250 m range gate during the ICICLE flight 20 (F20) conducted on 23 February 2019. As seen from this figure, the vertical and horizontal beam horizontal polarization reflectivities at X-band, where scattering by hydrometeors is generally in the Rayleigh regime, agree quite well. The corresponding mean X-band reflectivity difference is only about 1 dB. The vertical beam W-band reflectivites, however, are persistently greater than those at horizontal radar beam pointing by several decibel (~ 4dB). This is a manifestation of a non-Rayleigh effect of the backscatter enhancement from non-spherical particles with preferential orientation.

A similar effect of the vertical beam W-band reflectivity enhancement was observed during a number of field experiments with W-band radars (e.g., Matrosov 2012; 2019). The vertical radar beam reflectivities exceed those from horizontal beam measurements when ice particles mean dimensions in the horizontal plane are greater than those in the vertical plane. In other words, this non-Rayleigh reflectivity enhancement is observed when undisturbed particles are preferentially oriented with their major dimensions being near horizontal as dictated by aerodynamic forcing (even though some wobbling around this preferential orientation exists).

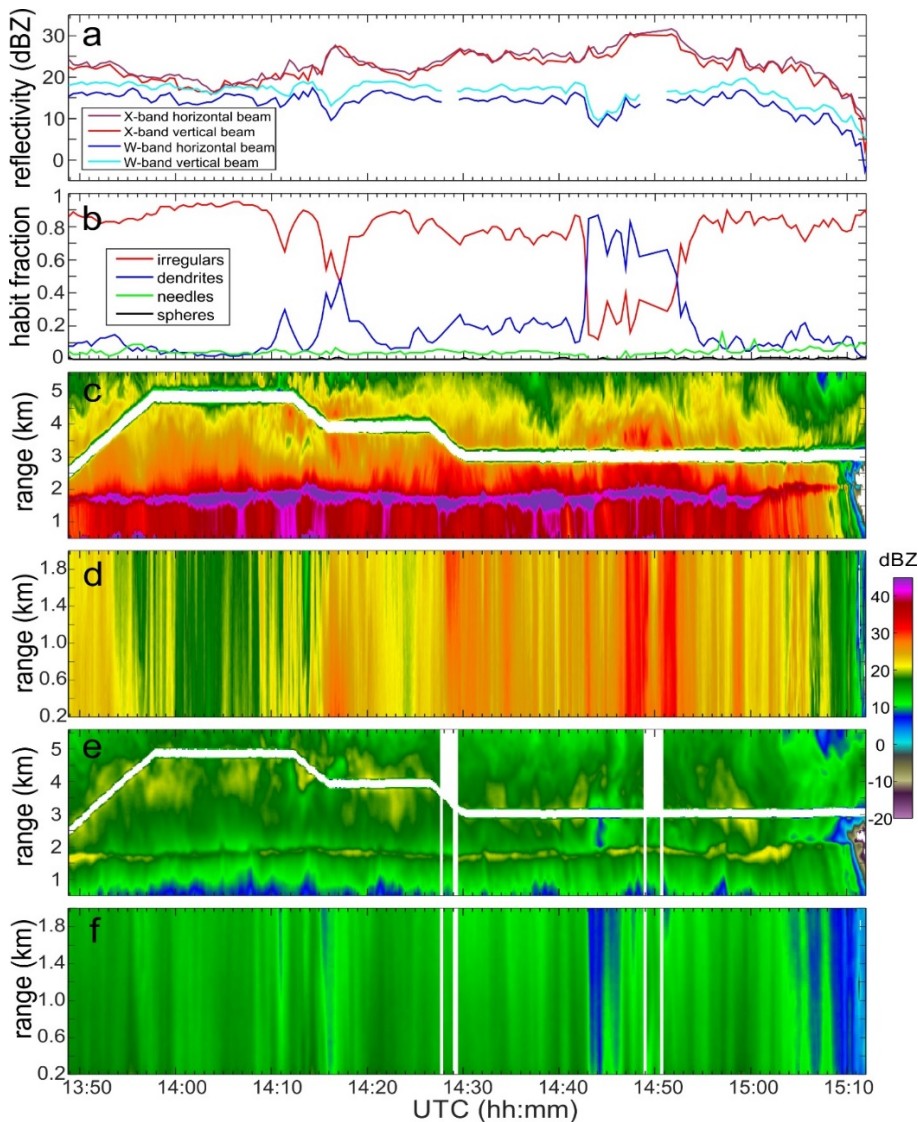

**Figure 2: F20 (a) time series of horizontal and vertical beam pointing measurements of NAWX reflectivities at a 250 m range gate, (b) estimates of relative fractions of different ice hydrometeor habits, (c) and (d) the X-band reflectivity time-height cross sections for vertical and horizontal beam pointing, (e) and (f) the W-band reflectivity time-height cross sections for vertical and horizontal beam pointing, respectively. Vertical white bands in (e) and (f) indicate no W-band data.**

The results of the 2D particle image recognition in Fig.2b show distinct ice hydrometeor regions with dendritic ice crystals embedded in the cloud filled by irregularly shaped ice particles. The Korolev and Sussman (2000) algorithm was applied in the recognition process using measurements from the OAP-2DC probe. Complete and partial hydrometeor images were included in the analysis. Inclusion of the partial images allowed an extension of the image recognition analysis to particles with sizes up to approximately 5 mm. The habit recognition was tuned in a way that aggregates of irregular particles and needles fall in the category of "irregular particles". However, the category "dendrites" included aggregates of dendrites. It is worth noting that for the ICICLE data set the occurrence of aggregates of needles was quite low.

As seen from Figs. 2a and 2b , the differences between the vertically and horizontally pointing W-band radar reflectivities are observed for all dominant particle habits. These differences depend on both the characteristic particle size and the degree of particle non-sphericity. Horizontal and slant no-data white bands indicating the aircraft trajectory in Figs. 2c and 2e correspond to the closest radar ranges. The aircraft was flying in precipitating ice conditions at heights between approximately 2.5 and 5 km with the radar bright band present at a height of about 1.7 km. The X-band bright band enhancement (Fig. 2c) is rather strong ($\sim$ 10 dB). The W-band bright band (Fig. 2e), which is caused, in part, by attenuation of high frequency radar signals by melting and liquid hydrometeors (Matrosov 2007; Sassen et al. 2007), is less pronounced ($\sim$3 dB).

For illustration purposes, Fig.3 shows images of typical populations of smaller ($D_v \approx 1.4$ mm) and larger ($D_v \approx 8$ mm) particles during ICICLE flight 20. The former was observed at around 14:03:28 UTC and the latter one corresponds to approximately14:48:26 UTC. As seen from Fig. 2a, the differences in X- and W-reflectivities around 14:03 UTC are rather small ($\sim 1 - 1.5$ dB), and are close to the DWR measurement uncertainty. This suggests that meaningful retrievals of $D_v$ from X-W-band dual wavelength measurements could be performed when this characteristic size exceeds about 1 mm.

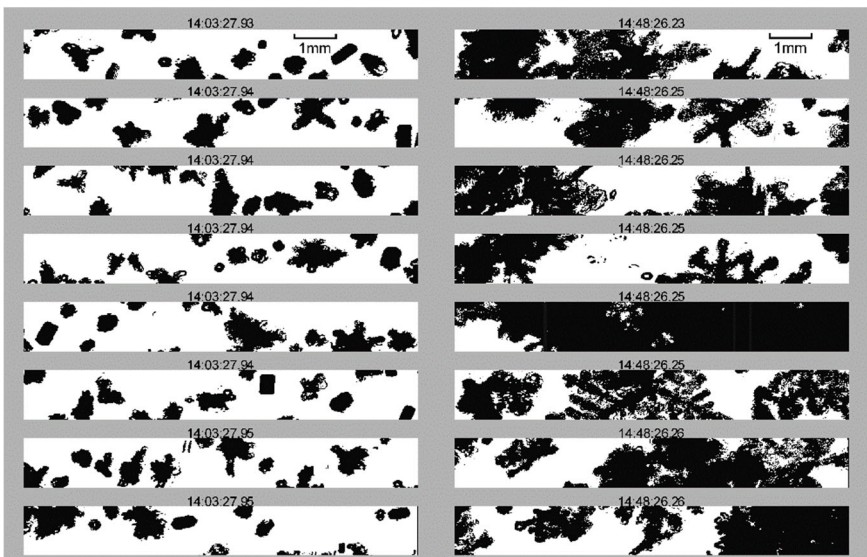

**Figure 3: Examples of particle 2D-S images (same scale) with $Dv \approx$ 1.4 mm and DWR $\approx$ 1.8 dB (left), and $D_v \approx$ 8 mm and DWR$\approx$16 dB (right). DWR data are for the horizontal radar beam measurements.**

Figure 4 shows scatter diagrams of $D_v$ and DWR for the horizontal and vertical radar beams measurements collected during the ICICLE flight 20. It appears that the best fit power-law approximations (red lines) of the $D_v$ - DWR relation works well when DWR is smaller than about 10-12 dB. Polynomial approximations (green lines) provide a better fit when DWR is greater than about 13 dB. Note also that practically all $D_v$ values for the F20 data are greater than approximately 1 mm.

One of the features of DWR, is that this ratio exhibits only a relatively weak dependence on particle bulk density
(Matrosov et al. 2019). Compared to other remote sensing approaches (e.g., Ryzhkov et al. 2018), this provides an important advantage when observing particle populations with habit dependent relations between particle mass and size (i.e., $m$-$D$ relations). These relations determine particle bulk densities and can vary depending on particle habits and presence of different microphysical processes (e.g., riming, aggregation).

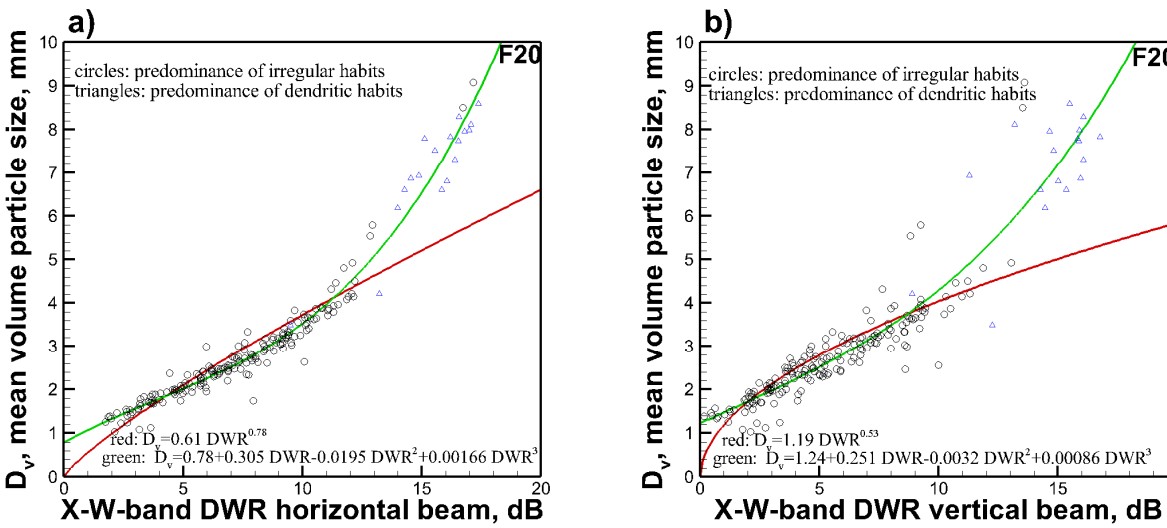

**Figure 4: Scatter diagrams of the mean volume particle size vs X-W-band DWR for (a) horizontal radar beam pointing and (b) vertical radar beam pointing during flight 20 measurements at the 250 m range gate. Best fit power law (red lines) and polynomial (green lines) approximations for all habits are also shown.**

As seen from Fig.4, the $D_v$ – DWR data scatter for the horizontal beam radar measurements (Fig.4a) is smaller
compared to the vertical beam (Fig.4b) (e.g., the horizontal and vertical beam $D_v$ RMSD values relative to the polynomial approximations are 0.62 and 0.72 mm, respectively). This is, in part, due to the fact that for the horizontal radar beam geometry of viewing, DWR is less susceptible to variations in ice particle shapes and orientations compared to that for the vertical radar beam measurements (Matrosov et al. 2019). It is worth noting that the data scatter (especially at horizontal radar beam viewing for the entire range of observed DWR) is rather modest despite the fact that the predominant particle habit through this cloud
segment was changing rather significantly (Fig. 2b). Though dendritic crystals produced, on average, higher DWR values due to generally larger sizes (e.g., Fig.3) during this flight, high DWR values were also observed when irregular particles were a dominant species.

**4 Statistical relations between PSD characteristic sizes and radar parameters**

**4.1 Differences between horizontal and vertical beam DWR measurements**

As mentioned in section 2, homogeneity of the cloud environment is an important factor for the analysis of the relations between the particle characteristic sizes and radar reflectivities. Since in this study the *in situ* microphysics and radar reflectivity measurements are separated by 250 m, cloud microphysics spatial inhomogeneities at scales $\Delta X < 250m$ would result in enhanced decorrelation and masking potential relations. As shown in Fig. 1, the clouds considered here were sufficiently homogeneous in the horizontal direction.

Figure 5 shows scatter plots of 250 m horizontal and vertical beam reflectivites. As seen from Fig. 5a, the vertical and horizontal beam X-band reflectivities, for which non-Rayleigh scattering effects are small, are still rather close. A modest bias of 1.2 dB can be attributed, in part, to the vertical anisotropy of cloud microphysical parameters related to various microphysical processes such as particle sedimentation, aggregation, diffusional growth and riming.

Unlike for the X-band, W-band reflectivities at the vertical beam pointing were significantly higher than those at the horizontal beam pointing (Fig. 5b vs Fig. 5a). This suggests that the non-Rayleigh reflectivity enhancement effects were more pronounced than those due to cloud inhomogeneity in the vertical direction. This leads to generally smaller vertical radar beam DWRs compared to those for horizontal radar beam DWRs for the same cloud microphysical parameters.

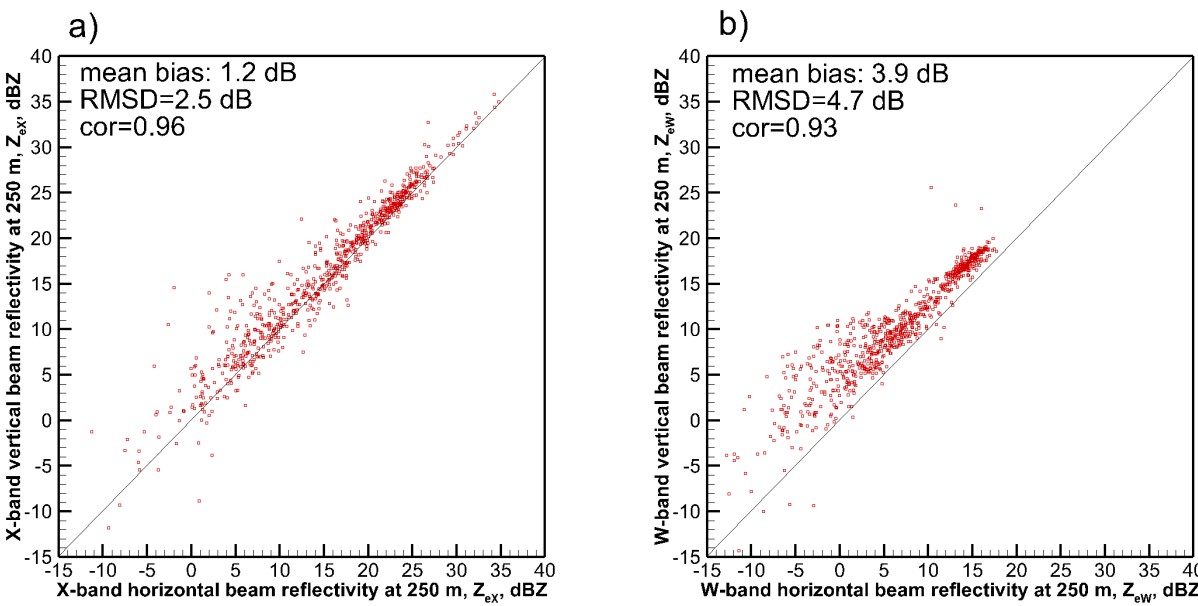

**Figure 5: Scatter plots of observed 250 m horizontal and vertical pointing reflectivities for (a) X-band and (b) W-band measurements.**

## 4.2 Relations between characteristic particle size and DWR

For all ICICLE flights considered in this study, Fig.6 shows frequency of occurrence scatter plots between the mean volume particle size $D_v$ and DWR for horizontal and vertical beam pointing. The bin sizes in this figure are 1 dB and 0.5 mm for DWR and $D_v$, respectively. The sample size consisted of ~800 $D_v$ – DWR pairs which represented 30 sec averages. The ice water content and temperature values in observed clouds varied in approximate ranges of 0.002– 2 g m$^{-3}$ and -2°C – -24°C, respectively.

As seen from Fig.6, DWR values for the same particle size are generally greater for the horizontal beam pointing data. There is an overall good correlation between $D_v$ and DWR. The $D_v$– DWR relation is generally more robust for horizontal beam measurements as the data scatter is smaller in Fig. 6a compared to Fig. 6b. The correlation coefficients between $D_v$ and DWR are 0.86 and 0.81 for the horizontal and vertical beam data, respectively. A better correlation for the horizontal beam DWR can be explained, in part, by the fact that hydrometeor backscatter for this beam pointing is less affected by particle shapes/habits (Matrosov et al. 2019).

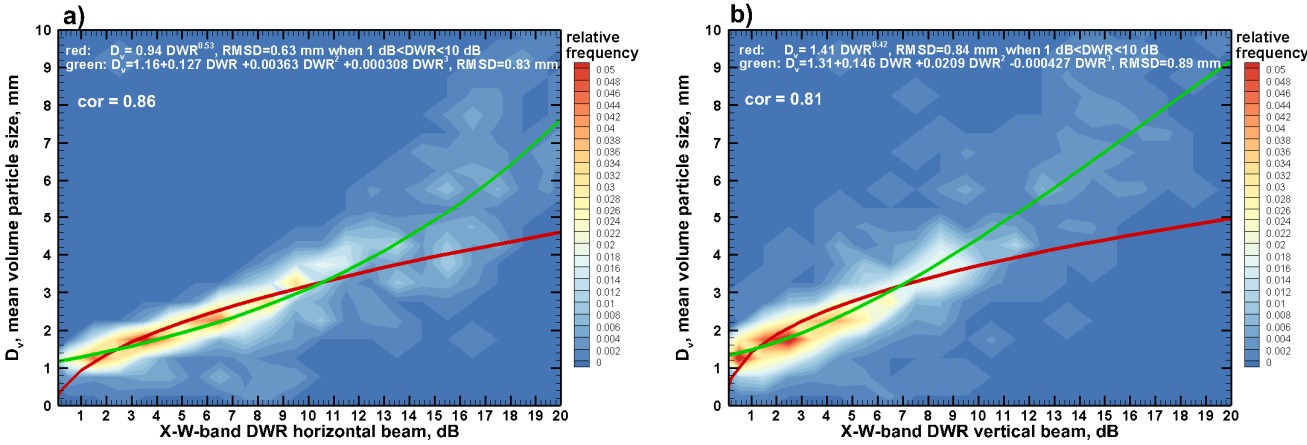

**Figure 6: Frequency scatter plots of mean volume PSD size, $D_v$, versus DWR observed with (a) and (c) horizontal and (b) and (d) vertical radar beam pointing.**

Best fit $D_v$– DWR approximations are also shown in Fig. 6. Unlike the polynomial fits, the power-law approximations do not adequately describe the $D_v$– DWR data for larger DWR values. These approximations, however, provide a better fit when DWR values are approximately in the 1 – 10 dB range, where the majority of the measurements are contained. The RMSD values of the best power-law fits shown in Figs. 6a and 6b (i.e., $D_v$=0.94 DWR$^{0.53}$, for horizontal beam measurements and $D_v$=1.41 DWR$^{0.42}$, for vertical beam measurements) are 0.64 mm and 0.84 mm, respectively if 1 dB < DWR < 10 dB. The corresponding RMSD values for the polynomial fits in Figs. 6a and 6b applied for entire observed range of DWR are 0.83 mm and 0.89 mm, respectively. If a DWR value of 1 dB is assumed as an uncertainty in X-W-band dual-wavelength measurements

then $D_v$ values of around 1 mm could be considered as the smallest characteristic size reliably retrievable from these measurement at horizontal radar beam pointing. Note also that even though the best fit approximations for the multiple flight data (Fig. 6) and those for one flight data (Fig. 4) differ, the characteristic size estimates from these approximations are quite close. For example, for the DWR range between 1 and 10 dB, a RMSD value describing a spread between $D_v$ estimates from the horizontal beam best power-law relations in Fig.4a and Fig.6a is only ~0.3 mm.

### 4.3 Relations between characteristic particle size and reflectivities

As seen from the data in Fig. 6a, the X-W-band DWR at horizontal radar beam pointing can be used for robust estimations of ice particle characteristic sizes describing PSDs if $D_v$ is greater than approximately 1 mm. Closely collocated dual-wavelength radar measurements, however, are not always available in many instances. Previous studies have shown that single frequency reflectivity measurements are also noticeably correlated with characteristic hydrometeor size if non-Rayleigh scattering effects are small (e.g., Matrosov 1999; Matrosov and Heysmfield 2017). Given this, it is instructive to evaluate statistical relations between single frequency NAWX reflectivity measurements and particle sizes using the ICICLE data.

Figure 7 shows $D_v$ - $Z_e$ frequency of occurrence scatter plots. The data are presented for the horizontal radar beam measurements, since those are less susceptible to the hydrometeor shape variability. As seen from Fig. 7a, there is a significant correlation (correlation coefficient ~0.72) between $D_v$ and $Z_{eX}$. The $D_v$ data scatter relative to the best fit power-law approximation of $D_v(\text{mm})=1.19Z_{eX}^{0.21}(\text{mm}^6\,\text{m}^{-3})$ is, however significantly larger than that for $D_v$-DWR relation (i.e., RMSD $\approx$ 1.3 mm in Fig. 7a versus RMSD $\approx$ 0.64 mm in Fig. 6a). The vertical radar beam data for X-band measurements (not shown) do not significantly differ from the horizontal beam measurements as reflectivities for both geometries of pointing are similar (Fig. 5a). The correlation between $D_v$ and W-band reflectivity is low (fig. 7b) as non-Rayleigh scattering diminishes backscatter dependence on particle size.

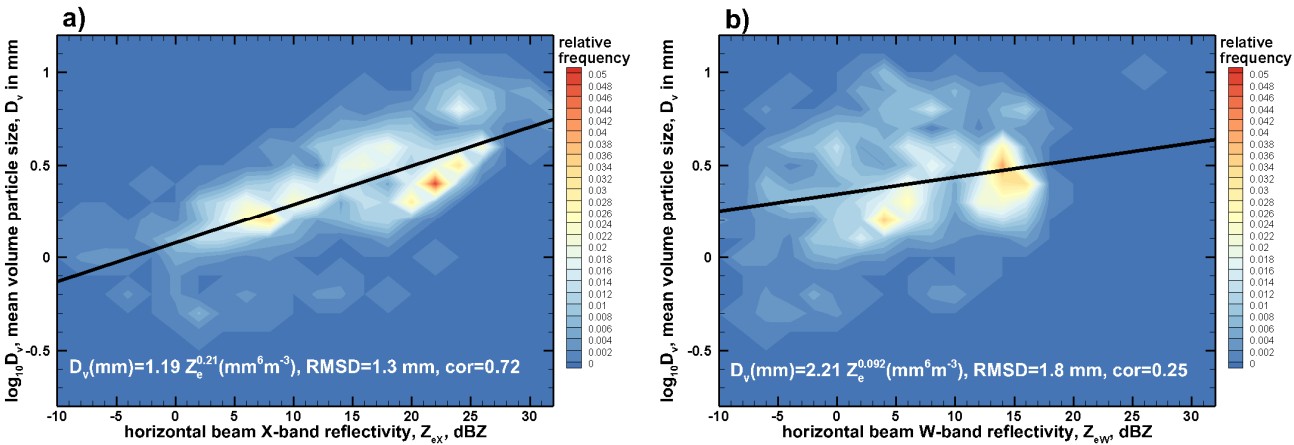

**Figure 7: Frequency scatter plots of mean volume PSD size, $D_v$, versus (a) X-band and (b) W-band reflectivity observed at horizontal beam pointing. Best fit power-law approximations and corresponding RMSD values are also shown.**

## 5. Relations among different definitions of PSD characteristic size

The relations among the characteristic size of ice hydrometeor PSDs and radar parameters discussed in previous sections were given in terms of mean volume particle size, $D_v$, defined as a ratio of fourth and third PSD moments (i.e., $M_4/M_3$). While $D_v$ is used in some remote sensing retrievals of ice hydrometeor parameters (e.g., Ryzhkov et al. 2018), a number of other characteristic particle sizes, which describe the entire PSD, are in common use in various remote sensing approaches and in different microphysical studies. The existence of ice microphysical data sets, where different definitions of ice particle characteristic size are used, necessitates establishing mean statistical relations among various definitions of PSD characteristic sizes. It is especially important when intercomparing results from different remote sensing approaches and model assumptions/parametrizations.

The mean volume diameter (MVD) is the characteristic size which is used in some applications (e.g., Schumann et al. 2011) including aircraft icing studies. While mean volume particle size, $D_v$, and MVD are similarly named, they are calculated differently. Using the PSD moment definition (2), MVD, which is defined using $0^{th}$ and $3^{rd}$ PSD moments, is expressed as

$$MVD = (M_3/M_0)^{1/3}. \tag{4}$$

Figure 8a shows a frequency scatter plot between MVD and $D_v$ as calculated from the ICICLE *in situ* data set used in this study. As seen from this figure, MVD values are generally smaller than $D_v$ values by approximately a factor of 2 on average.

Another widely used characteristic particle size describing PSDs is the median volume size, $D_0$. For hydrometeors of similar shapes and nontrancated gamma-function size distributions, which are often used to approximate observational PSDs, the theoretical relation between $D_0$ and $D_v$ is

$$D_0 = (3.67+\mu)(4+\mu)^{-1} D_v, \tag{5}$$

where $\mu$ is the order of the gamma-function. It can be seen from (5) that for exponential distributions ($\mu=0$), $D_0$ and $D_v$ differ by less than approximately 10%. This difference diminishes even further for higher orders of gamma-function PSDs.

A median mass size, $D_m$, is sometimes used instead of $D_0$ for describing PSDs. For particles of the constant bulk density, $D_m \approx D_0$. When bulk density changes with size, which is the case for ice hydrometeors, $D_m$ also depends on the individual particle mass – size relations (i.e., $m$-$D$ relations). Figure 8b shows the relation between $D_m$ and $D_v$ obtained using the $m$-$D$ relation (i.e., $m=0.00338\,D^{1.9}$, cgs units) for collected ICICLE PSDs. This relation was found as a result of comparing the PSD-to-mass calculations and bulk ice mass content estimates measured by the isokinetic probe (IKP, Davison et al. 2011) in ice clouds. This allowed finding the $m$-$D$ relation coefficients that provide best matching of the ice mass calculated from PSDs and that measured directly by the IKP. As seen in Fig.8b, $D_m$ is, on average, about 70% of $D_v$.

Sometimes PSDs are characterized by the mean particle size (e.g., Shupe et al. 2006), which is defined as the ratio of the $1^{st}$ and $0^{th}$ PSD moments:

$$D_{mean} = M_1/M_0 . \tag{6}$$

The statistical relation between $D_v$ and $D_{mean}$ is shown in Fig. 8c. $D_{mean}$ is generally significantly smaller than $D_v$.

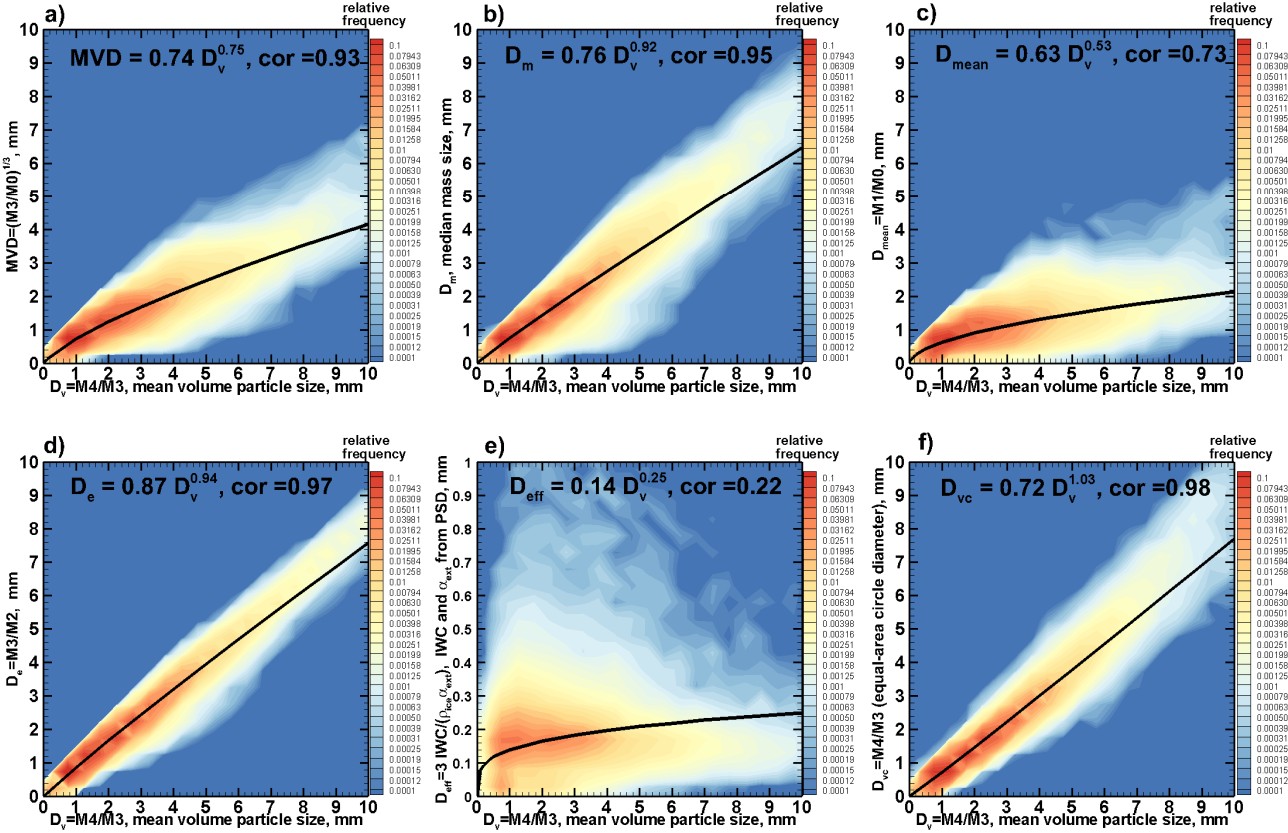

**Figure 8: Observationally-based statistical relations between (a) MDV and $D_v$, (b) $D_m$ and $D_v$, (c) $D_{mean}$ and $D_v$, (d) $D_e$ and $D_v$, (e) $D_{eff}$ and $D_v$, and (f) $D_{vc}$ and $D_v$ according to the ICICLE *in situ* sampling data. Corresponding correlation coefficients are also shown. Note that the Y-axis scale in frame 8e is different from other frames.**

The effective particle size (diameter), $D_e$, is often used in optical remote sensing and model parameterizations. It is frequently defined as the ratio of the 3rd and 2nd PSD moments (e.g., Schumann et al. 2011):

$$D_e = M_3/M_2 \tag{7}$$

Quite often, however, the effective radius, (i.e., $r_e = D_e/2$) is used instead of the effective diameter (McFarquhar and Heymsfield 320   1998).

The statistical relation between $D_e$ and $D_v$ obtained using ICICLE microphysical data is shown in Fig. 8d. It can be seen that, if the effective size is expressed in physical particle sizes (i.e., as a major dimension of a two-dimensional particle projection), then on average $D_e$ is about $0.8D_v$, though there is some data scatter around the best-fit approximation. For particles of similar habits and size-independent bulk density, ρ, the effective size can be expressed in terms of liquid or ice cloud water 325   content (WC) and visible extinction coefficient, $\alpha_e$, (e.g., Mitchell et al. 2011):

$$D_e = 3 \, WC \, /(\rho \, \alpha_e) \,. \tag{8}$$

For spherical water drops, the definitions (7) and (8) are equivalent and they provide the effective size in terms of particle physical dimensions. If (8) is applied to ice hydrometeors and the density of the solid ice is used (i.e., $\rho=\rho_i=0.916$ g cm$^{-3}$), it becomes:

$$D_{eff} = 3 \, IWC \, /(\rho_i \, \alpha_e) \,, \tag{9}$$

where IWC is ice water content and the notation $D_{eff}$ (rather than $D_e$) underlines that the effective size here is derived not in terms of the PSD moments using physical dimensions of observed hydrometeors but rather from measurements or estimates of the extinction coefficient and IWC ($D_{eff} \neq M3/M2$). This definition of $D_{eff}$ is used in various applications (McFarquhar and Heymsfield 1998; Mitchell et al. 2011).

The ICICLE microphysical data set allows for evaluating the relation between $D_{eff}$ defined by bulk quantities and PSD characteristic sizes expressed in terms of particle physical dimensions. The $D_{eff} - D_v$ frequency of occurrence scatter plot as derived from the ICICLE data set is shown in Fig. 8e. When calculating $D_{eff}$, the IWC data were obtained using the $m=0.00338 \, D^{1.9}$ relation with observed composite PSDs and the extinction coefficient was approximated using estimates of the total projected area of all particles in the size distribution (Korolev et al. 2014). $D_{eff}$ depends on the assumed $m$–$D$ relation

(i.e., it is proportional to the prefactor in this relation).

      It can be seen from Fig. 8e that $D_{eff}$ is significantly smaller than $D_v$. The majority of $D_{eff}$ values are under 0.25 mm regardless of the $D_v$ magnitude. Unlike for other characteristic sizes, a power-law approximation does not provide a robust fit for the $D_{eff}$-$D_v$ relation, and the correlation coefficient between these two characteristic ice hydrometeor sizes is very low (cor = 0.22). Such a weak dependence of $D_{eff}$ on $D_v$ can be explained by the fact that both particle mass (hence IWC in the numerator

of (9)) and their cross-sectional area (hence the extinction coefficient in the denominator of (9)) are both approximately proportional to particle physical size squared (e.g., Mitchell 1996), so their ratio does not significantly depend on particle physical sizes.

      The characteristic particle sizes given by (3) – (7) and shown in Figs. 8a-8d were calculated using moments of PSD given in terms of the major particle dimensions/projections as inferred from 2D probes. Sometimes, particle dimensions are

350 given in terms of diameters of the circles that have the same projection area as 2D particle images (e.g., McFarquhar and Heymsfield 1998). Figure 8f shows a $D_v - D_{vc}$ frequency scatter plot where $D_v$ and $D_{vc}$ are calculated as the $M4/M3$ ratio in terms of particle major dimensions and diameters of the equal-area circles, respectively. It can be seen that $D_{vc}$ values are, on average, smaller than those of $D_v$ by about 20-30%. Power-law approximations of the statistical relations among different characteristic sizes of ice hydrometeors, which are shown in Fig.8, can be used to convert (at least in a mean sense) the $D_v -$

DWR relations discussed in section 5 to such relations given in terms of various other definitions of particle characteristic size describing an entire PSD.

Correlations between most hydrometeor PSD characteristic sizes expressed in terms particle physical dimensions are quite high (e.g., Figs. 8a, 8b, 8d). Very high is also the correlation between the characteristic size expressed in terms of the maximal 2D projection and the one in terms of the diameter of equal-area circle (Fig. 8f). Somewhat smaller but still significant

is the correlation coefficient between $D_v$ and $D_{mean}$. Statistical relation shown in Fig. 8 can facilitate meaningful comparisons of characteristic particle sizes from different retrievals as well as from different microphysical parameterisations in climate and weather models. Approximate relations between DWR and the characteristic particle sizes other than $D_v$ can be readily obtained from the $D_v$-DWR relations (Fig. 6) by expressing these sizes in terms of $D_v$ using best fit power-laws shown in Fig.8. Correlation coefficients between the horizontal radar beam DWR and MVD, $D_m$, $D_e$ and $D_{vc}$ are approximately in the 0.82–

0.86 range, which is close to the correlation between DWR and $D_v$. DWR and $D_{mean}$ are less correlated (cor≈0.6), which is mostly due to the lower correlation between $D_{mean}$ and other characteristic sizes (Fig. 8c). No meaningful correlation exists between DWR and $D_{eff}$. The correlation coefficients between the vertical beam DWR and characteristic particle sizes are by about 0.05 smaller compared to those for the horizontal beam DWR.

**6. Conclusions**

Radar and microphysical data sets collected during the ICICLE project were used to quantitatively evaluate relations between X-W-band dual-wavelength airborne radar measurements and characteristic sizes of ice hydrometeor populations. A close collocation of the radar and microphysical sampling measurements allowed for robust comparisons of microphysical and radar data. To minimize effects of cloud microstructure inhomogeneity, the consideration was limited to ice cloud regions

where radar reflectivity was assessed as spatially homogeneous within ranges up to several hundred meters. The data considered here were also limited to the regions where radar echoes were dominated by ice hydrometeors. In the selected cloud regions, the mean volume particle size $D_v$ varied from a few hundred micrometers to about 1 cm and ice particle shapes were presented by a variety of major ice habits including dendrites, needles and irregulars.

Radar reflectivities observed with horizontal and vertical radar beam pointing did not exhibit significant differences at

the X-band radar frequency, where hydrometeor backscatter is mostly within the Rayleigh scattering regime. In this scattering regime, this generally agrees with the horizontally oriented spheroidal particle model (e.g., Bohren and Huffman 1983). Unlike at X-band, reflectivity values at W-band for the vertical radar beam pointing were consistently higher than those for the horizontal radar beam measurements due to zenith/nadir reflectivity enhancement, which is explained by non-Rayleigh scattering by non-spherical particles with preferential orientation. Due to this enhancement, DWR values at vertical incidence

were, on average, about 4 dB lower than those for the horizontal beam measurements.

The X-W-band dual-wavelength ratio at horizontal beam pointing was found to provide a robust tool for retrieving the characteristic particle size if $D_v > 1$ mm. The influence of particle non-sphericity and orientation is minimized at this geometry of viewing. A measure of uncertainty of the retrieval [as estimated using the RMSD from the mean ICICLE best fit approximation of $D_v(mm)=0.94 \ DWR^{0.53}$ (dB)] is about 0.64 mm if 1 dB<DWR<10 dB. While the power-law approximations

work better for this DWR range, the $D_v$ – DWR relation for higher dual-wavelength ratio values is better approximated by a

polynomial function. The $D_v$-DWR relation for the vertical beam measurements exhibits higher variability (compared to horizontal beam pointing) due to W-band zenith/nadir reflectivity enhancements, which are particle habit and orientation dependent.

DWR-based particle characteristic size retrievals generally do not require the absolute calibration of radar reflectivities. A relative calibration of the dual-wavelength signals can be potentially performed when observing cloud regions with smaller crystal populations which provide the Rayleigh regime of scattering at both radar frequencies. Another advantage of the DWR approach for sizing ice particle populations include a relatively little dependence of dual-wavelength radar signal on particle bulk density.

Single frequency reflectivity measurements at X-band are also informative on the characteristic particle size. Correlation coefficients between $Z_{eX}$ and $D_v$, however, is noticeably lower than those between DWR and $D_v$ (~ 0.72 versus ~0.86, respectively for the horizontal beam pointing). The RMSD value for $D_v$, - $Z_{eX}$ relation is ~1.3 mm. Reflectivity-based estimates of ice particle characteristic sizes, however, can be also be obtained for smaller values of $D_v$, while the X-W band DWR-based hydrometeor sizing approach is generally viable for $D_v > 1$ mm.

There are different characteristic sizes used to describe particle populations. For a same PSD, various characteristic sizes differ in magnitude. The mean volume size, $D_v$, is usually larger, than other commonly used characteristic sizes describing PSDs such as, the median mass size, the mean size, the effective size, the mean volume diameter, etc. Characteristic size nomenclature/definition differences need to be accounted for when comparing data from different retrieval approaches and model studies. Approximate relations among characteristic sizes other than $D_v$ and DWR can be readily obtained from $D_v$ - DWR relations and relations between these other sizes and $D_v$ (Fig. 8).

Overall, the results of this study indicate the robustness of the DWR radar approach for inferring characteristic sizes of ice hydrometeors. Since atmospheric ice particles are generally nonspherical and are not randomly oriented, the influences of the geometry of viewing are important for dual-wavelength methods and, also for triple-wavelength approaches, which predominantly use vertical beam measurements. Measurements with near horizontal radar beam pointing are better suited for size retrievals as they are less susceptible to the effects of particle non-sphericity and preferential orientation, so characteristic size retrievals for this measurement geometry have smaller uncertainties. Future enhancements of multi-wavelength radar remote sensing methods for retrievals of ice hydrometeor microphysical parameters need to account for realistic orientations of particles. The ICICLE radar measurements can provide a valuable data set for testing different computational approaches of calculating ice hydrometeor scattering properties.

Besides the NAWX, X-W- band radar measurements are available from a number of airborne platforms (e.g., Heymsfield et al. 2016) and ground-based sites including the U.S. Department of Energy's Atmospheric Radiation Measurement (ARM) facilities (Kollias et al. 2014). The suggested here relations between DWR and the characteristic particle size can be used with measurements from these radar data. Attenuation effects (especially those at W-band band), however, need to be accounted for if radar measurements are performed at longer ranges. The results of this study could also be potentially useful for the dual-wavelength radar measurements where lower radar frequencies such those at S-band or C-band (i.e., ~3 GHz or ~ 5.5 GHz,

respectively) are used instead of the X-band frequency. Combinations of S- and W-band frequency radar measurements could be available from spatially matching spaceborne *CloudSat* W-band reflectivities with those from the U.S. Next Generation Weather Radar (NEXRAD) ground-based S-band systems (e.g., Matrosov 2010).

*Data availability*. The ICICLE data used in this study is available from the NCAR/UCAR Earth Observing Laboratory (EOL) archive: W-band radar data https://doi.org/10.26023/PBVG-0S4X-3D05; X-band radar data
https://doi.org/10.26023/KAGA-JH3J-0Y06; hydrometeor microphysics data https://doi.org/10.26023/DREN-VTHA-0N0E; https://doi.org/10.26023/7BZP-NGPY-WG0B and https://doi.org/10.26023/R6A2-G92Q-CF0S.

*Author contributions*. Conceptualization of the remote sensing approach was performed by SYM. Data quality control and processing were performed by AK (hydrometeor microphysics data) and MW and CN (radar data). Data interpretation,
analysis and writing were shared by SYM and AK.

*Competing interests*. The authors declare that they have no conflict of interest.

*Acknowledgements. This research was supported, in part, the US Department of Energy (DOE) Atmospheric Systems Research (ASR) program projects DE-SC0022163. The ICICLE field campaign was funded by the Federal Aviation Administration (FAA). The views expressed are those of the authors and do not necessarily represent the official policy or*
440 *position of the FAA.*

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
