# Peer review of "Sizing ice hydrometeor populations using dual-wavelength radar ratio"

_Atmospheric Measurement Techniques, 2022_

## Author Comment (AC2)

**Scale: -10 <Z<44 dBZ**

[Figure]

**Scale: -5 <Z<44 dBZ**

[Figure]

**Scale -20 <Z< 44dBZ**

---

## Author Response (AR1)

Responses to the reviewer 1 comments

We thank the reviewer for constructive comments. Below are the responses to these comments.

Minor comment 1: Manuscript title: Consider adding which types of clouds this retrieval applies to or include the ICICLE campaign as these results were not extensively tested on other cloud/system types.
Response: To address this comment we now specifically mention in the abstract that ICICLE measurements were conducted in winter-time midlatitude frontal clouds. Additionally, in the main text we mention that the ice water content and temperature values in observed clouds varied in approximate ranges of 0.002– 2 gm$^{-3}$ and -2$^o$C –  -24$^o$C, respectively.

Minor comment 2: L90: More details on the microphysics are needed. What combinations of probes were used (e.g., PIP vs. HVPS) for the composite size distributions? What size ranges were used for the 2D-S and PIP/HVPS when combining the distributions? How were the probes oriented?
Response: The PIP particle spectrometer was used as a backup for the HVPS. Since all composite PSDs were calculated using HVPS, the statement about PIP (line 85-86) was excluded from the text to avoid confusion. To address the comment about the calculation of composite DSDs the following text has been added at the end of section 2.1: "Calculations of composite PSDs for ice cloud segments employed 2D-S and HVPS measurements in the following size subranges: 40µm to 670µm (at 10µm pixel resolution) and 750µm to 3.84cm (at 150µm pixel resolution), respectively." The HVPS was mounted to measure vertical projection of cloud particles, whereas the 2D-S probe allowed measurements of both vertical and horizontal projections of ice particles. Composite 2DS-HVPS PSDs were calculated for particles having the same projections. However, occasionally in case of malfunctioning of the 2D-S vertical channel, the measurements of the horizontal 2D-S channel were used for calculation of composite PSDs.

Minor comment 3: 2b and L156: Can you comment on whether only 2D-S images were used in the habit classification? I assume this is the case. If so, you should mention that the true habit breakdown may be different than shown since larger crystals may constitute aggregates from columns or planar crystals.
Response: The image recognition processing to identify habits of cloud particles (Korolev and Susman, 2000) was applied to the measurements of the OAP-2DC probe (Knollenberg, 1981). Because of its more coarse pixel resolution (50µm) the OAP-2DC particle imagery provided a better results of particle habit classification compared to the 2D-S. Particle images of 2D-S, because of its high pixel resolution (10µm), are subject of diffraction and out-of-focus effects. These effects degrade results of identification of particle shapes, and therefore, the 2D-S measurements were not used for ice particle habit recognition. The habit recognition was tuned that way that aggregates of irregular particles and needles fall in the category of "irregular particles". However, the category "dendrites" included aggregates of dendrites. Both complete and partial (i.e., larger size particles) particle images were used for the habit classification. Inclusion of partial images in the processing enabled an extension of the image recognition analysis to about 5 mm. The information given above was included in the revision.

Minor comment 4: L176 & Fig. 4: "…provide a better fit…" could probably use a statistic quantifying this agreement.
Response: This statement (Fig. 4 is for the Flight 20 data) is clarified in the revised manuscript. It is pointed out now that the polynomial approximation is better fitted for DWR greater than about 10-13

dB. Note also that RMSD values for power-law and polynomial best fits are given when discussing data from multiple flights given in Fig. 6.

Minor comment 5: L220: Specify the DWR and Dv bin increments used to generate Fig. 6.
Response: The bin sizes in this figure are 1 dB and 0.5 mm for DWR and $D_v$, respectively.   The statement about it is added in the revised manuscript.

Minor comment 6: L292: Can you elaborate on the prefactor in the Brown and Francis m-D relationship? The value listed doesn't seem to reflect the a = 7.38 x 10^-11 g µm^-1.9 in their study after converting to cgs units. Further, their study used a Dmean definition for particle size (Hogan et al. 2012; DOI: 10.1175/JAMC-D-11-074.1), while this study uses a Dmax definition (L329).
Response: Indeed, the cgs unit prefactor coefficient 0.0033787 (note there was a typo in the original manuscript) in the m-D relation used in our study corresponds to a larger coefficient compared to the one in the B&F study (the exponent coefficient of 1.9 is the same). The prefactor coefficient used in our study was found as a result of comparisons of the PSD-to-mass calculations and bulk IWC measured by the isokinetic probe (IKP, Davison et al. 2011). This allowed finding coefficients $a$ and $b$ in the power-law m-D relation (where D is a larger particle projection, unlike in the B&F study) that provide best matching of the IWC calculated from PSDs and that measured directly by the IKP. The corresponding explanation is given in the revised manuscript. Note that the choice of the coefficient in the m-D relation only affects Deff (eq. 9) but not other characteristic sizes considered here. We are sorry for the confusion caused by the original text.

Minor comment 7: Fig. 8 nicely demonstrated how the different definitions of characteristic size relate to Dv. The end of Sec. 4 alludes to how this analysis "can facilitate meaningful comparisons of characteristic particle sizes from different retrievals" but falls short of demonstrating this link to the DWR measurements. To extend upon the Dv-DWR fits shown in Fig. 4, have the authors thought about adding a figure showing either a power law or a polynomial fit for each combination of characteristic size – DWR?
Response: We added a following statement in the revised manuscript: "approximate relations between the characteristic particle sizes other than $D_v$ and DWR can be readily obtained from the $D_v$-DWR relations (Fig. 6) by expressing these sizes in terms of $D_v$ using best fit power-laws shown in Fig. 8". For example, to obtain a relation between MVD and DWR, relations MVD=0.74 $D_v^{0.75}$ (Fig. 8a) and $D_v$=0.94 DWR$^{0.53}$ (Fig. 6a) can be combined to obtain MVD=0.74(0.94DWR$^{0.53}$)$^{0.75}$=0.71DWR$^{0.40}$.

Technical correction 1: L103: References should be moved to be in line with the rest of the sentence rather than in parentheses.
Response: Fixed.

Technical correction 2: L208: process -> processes
Response: corrected.
Technical correction 3: L315: MacFarquhar -> McFarquhar
Response: corrected.
Technical correction 4: L322: From -> from
Response: corrected.

Responses to the reviewer 2 comments
We thank the reviewer for constructive comments. Below are the responses to these comments.

*General comment 1: How general are the observed correlations?*
From the perspective of a potential user whose aim is to retrieve characteristic sizes from dual-frequency radar measurements, some information may be missing. It would be relevant to include more information on the data that were collected. For example, how many data points were included - how long were the time periods used (after the preprocessing steps which include e.g. removing mixed-phase conditions) to compute the coefficients of Fig 6? What were the flight conditions like, in terms of (for example) altitude, temperature, total water content...? Similarly, the authors mention the presence of various particle habits, but this could be expanded (e.g. indicate the classes of particles present in the entire dataset used, mention the presence of riming if any...). All this could help the reader get an idea of how general the derived relations are, and how they might hold in various environments.

Response: In the revised paper we mention that the sample size consisted of ~800 $D_v$ – DWR pairs which represented 30 sec averages. The ice water content and temperature values in observed clouds varied in approximate ranges of 0.002 – 2 gm$^{-3}$ and -2°C – -24°C, respectively. Aircraft flight altitudes varied between about 2 and 5 km. Additionally, we now mention in the abstract that sampled precipitating clouds were generally winter-time midlatitude frontal clouds. We also mention that particles were mostly of irregular type with pockets of dendritic type crystals. The fraction of needle-type particles was generally very low. There were no cloud segments with significant riming. Though, riming is not expected to provide high impact as it changes particle densities, but densities do not significantly affect DWR values (e.g., Matrosov et al. 2019).

*General comment 2. Could a little more be said about the vertical beam results?*
The authors point out that the DWR-Dv relations are more robust when using the horizontal beams. Unfortunately, most long-term radar observations are either space-borne or ground based and thus rely on vertical beams; hence, I wonder if a little more could be said on the DWR-Dv relations with vertical beams. Is the spread completely "random", or perhaps could it be easily related to varying particle habits, aspect ratios...? for example a suggestion could be to color-code the scatter plot e.g. on Fig. 4 with particle type or aspect ratio.

Response: As suggested, a color-coding was added in Fig. 4. The spread in DWR-Dv points around the best fit for the horizontal radar beam is smaller than that for the vertical beam regardless of the predominant habit, even though large dendritic crystals (Fig. 4) often (but not always) produced higher DWR values.

*General comment 3. How do the results compare to model studies?*
While I understand and value that the paper's focus is an observational study and not a modeling work, I believe a bridge between the two approaches would be of high interest. Indeed, there have been numerous model-oriented investigations of multi-frequency radar measurements in the past decade (including but not limited to Kneifel 2011, 2015, Leinonen 2012, Ori 2015, Mason 2019, Oue 2021). Even

without studying in-depth the accuracy of various models/parameterizations, perhaps some literature results could be used almost as is for comparison (e.g. similar as Matrosov 2019, but with the appropriate frequencies?).

Response: Yes, there have been numerous model-oriented studies of dual- and triple frequency radar ratios. Most of these studies, however, assume random orientations of ice particles. For this assumption, there is no difference between measurements with vertical and horizontal beam pointing. We believe, that one important outcome of our study is providing an observational evidence that there are important DWR signature differences between different geometries of radar beam pointing which can only be explained by the fact that non-spherical particles are preferentially oriented. To a certain extent, a simple oriented spheroidal model can explain the fact that DWR(hor_beam) > DWR(ver_beam). However, given a multitude of different more sophisticated computational models/approaches and a general lack of easily available (in literature) simulated DWR-particle size relations for the W-X band frequency pair, it is not very feasible to provide a meaningful comparison between observational and modeled DWR (from different computational approaches) within a framework of the current study. Such a comparison would be highly desirable in future and it should involve modeler collaborators. We included some thoughts on this in the conclusion section of the revised manuscript.

Specific comment: Sect. 2.1 How were the radars calibrated?

Response: Relative calibration of DWR is achieved by assuming that DWR is 0 dB for the regions of low reflectivity (i.e., small crystals). Absolute calibration is made using clear-air observations of the water surface backscatter cross section (Li et al., 2005). Calibration for other NAWX antennas is done by comparing measurements between antenna ports. More details (and references) on calibration are described by a study of Nguyen et al. (2019), which is referenced in the manuscript.

Specific comment: l. 107 How is the liquid drop mean volume diameter computed (and the LWC)? With the CDP probes?

Response: Liquid versus ice hydrometeor type identification was performed based on the analysis of measurements of a combination of the Rosemount Icing Detector (RID, Mazin et al. 2001), the particle scattering probes (CDP, FCDP), the Cloud Particle Imager (CPI, Lawson et al. 2001) and the 2D-S probe. The choice of the probes was done based on proximity of the LWC measured by the probes to the LWC measured by the RID. The analysis of the RID and scattering probes measurements was primarily used to identify the presence of small liquid cloud droplets. Liquid MVD and LWC values for liquid drop populations were calculated for particles which were identified as liquid drops. We included the relevant explanation in the revised manuscript.

Specific comment: Fig. 2: I recommend the authors re-do this figure. It is 1) very stretched, and 2) the colorbar of the radar data (plots c - f) is not adjusted (it goes down to -20 dBZ but there are no such low values). This makes the reading quite difficult.

Response: We changed the aspect ratio of this figure. Now it looks less starched. To address the reviwer comment, we also attempted different color bars, including -20 dBZ to 44 dBZ, -10 dBZ to 44 dBZ, and -5 dBZ to 44 dBZ. Examples with the use of these color bars are shown below. We think that the – 20 dBZ to 44 dBZ color bar presents a better overall view as it provides a better color coverage for X-band radar data and also it provides an additional yellow hue (compared to other color bars) for the W-band radar data.

**Scale: -10 <Z<44 dBZ**

[Figure]

**Scale: -5 <Z<44 dBZ**

[Figure]

**Scale -20 <Z< 44dBZ**

[Figure]

Specific comment: l. 156 Which probes were used for the hydrometeor classification? Are all the size ranges included?

Response: The image recognition processing to identify habits of cloud particles (Korolev and Susman, 2000) was applied to the measurements of the OAP-2DC probe (Knollenberg, 1981) which has a 50 μm – 1.6 mm size range (with a 50 μm image resolution). Complete and partial images were included in the analysis. Inclusion of partial images in the analysis allowed an extension of the image recognition analysis to particles with sizes up to about 5 mm. This information was included in the revised manuscript.

Specific comment: Fig. 3: Please indicate the scale on the images. It could be relevant to also show the HVPS images at the corresponding time steps: on the right panels, the particles are large and the 2DS images are not very telling.

Response: We included the scale in Fig. 3 following the review comment. However, we consider adding HVPS images redundant here since they do not significantly add to the information on particle habits.

Specific comment: Fig. 8: It would be interesting to also have some results on the observed correlations between DWR and the other characteristic sizes. Is Dv the characteristic size which correlates the best with DWR?

Response: Correlation coefficients between the horizontal radar beam DWR and MVD, $D_m$, $D_e$ and $D_{vc}$ are approximately in the 0.82– 0.86 range, which is close to the correlation between DWR and $D_v$. DWR and $D_{mean}$ are less correlated (cor≈0.6), which is mostly due to the lower correlation between $D_{mean}$ and other characteristic sizes (Fig. 8c). No meaningful correlation exists between DWR and $D_{eff}$. The correlation coefficients between the vertical beam DWR and characteristic particle sizes are by about 0.05 smaller compared to those for the horizontal beam DWR. This information was added in the revised manuscript.

Specific comment: l. 289-293. The mass-size relation of ice hydrometeors can vary significantly depending on the particle population (e.g. Mason 2018, Leinonen 2021) . Why was this specific relation chosen? What would be the impact of changing it? Perhaps distinguishing particle types would cause some differences.

Response: The exponent of the m-D relation is the same as in the Brown and Frances (1995) m-D relation which is widely used in the community. The prefactor coefficient in this relation used in our study was found as a result of comparisons of the PSD-to-mass calculations and bulk IWC measured by the isokinetic probe (IKP, Davison et al. 2011). This allowed finding coefficients $a$ and $b$ in the power-law m-D relation (where D is a larger particle projection) that provide best matching of the IWC calculated from PSDs and that measured directly by the IKP. The corresponding explanation is given now in the revised manuscript. Note that the choice of the coefficient in the m-D relation only affects Deff (eq. 9) but not other characteristic sizes considered here. The DWR-Dv relations are not affected by the choice of the m-D relation since the m-D relation is not used for calculations of characteristic sizes other than Deff.

Specific comment: l. 319: Same question – why this relation? was it found appropriate in the ICICLE dataset? Otherwise, would the results differ significantly with another relation? Did the ICICLE payload not include some direct TWC / IWC measurements (perhaps these were not available)?

Response: There were two Nevzorov bulk probes installed during the ICICLE flights. IWC estimates from these probes differ by about 13% on average. We decided to use PSD-based IWC estimates (see the response to the previous comment) for calculating $D_{eff}$ using equation 9 (i.e., this the only equation where IWC is used). This decision was also dictated that by the fact that extinction coefficient in this equation (i.e., $\alpha_e$) was also estimated using observed PSD. This will result in error compensation and is expected to provide a better estimate of $D_{eff}$. We make a note in the revised manuscript that $D_{eff}$ values are dependent on the choice of the m-D relations (e.g., $D_{eff}$ is proportional to the prefactor in this relation).

Technical comment: l. 46 "remote sensing algorithms" → "remote sensing retrievals"?
Response: changed as suggested.

Technical comment: l. 77 "from this regime occurring" → "occur"
Response: changed as suggested.

Technical comment: l. 77 "which sizes" → "whose sizes"
Response: changed as suggested.

Technical comment: l. 164 "Sassen et a" → "Sassen et al."
Response: changed as suggested.

Technical comment: l. 166 "the later one" → "the latter one"
Response: changed as suggested.

Technical comment: l.230 "The RMSD values of observed Dv values from the best power-law fits...": perhaps simply "The RMSD of the best power-law fits"
Response: changed as suggested.

Technical comment: l. 289 "is sometimes used instead D0" → "is sometimes used instead of D0"
Response: changed as suggested.

Technical comment: l. 303 "The ICICLE microphysical data based statistical relation" → a little unclear
Response: Changed to "The statistical relation obtained suing ICICLE microphysical data.

Technical comment: l. 320 two spaces after "m-D relation"
Response: fixed.

---

## Author Response (AR2)

Responses to the reviewer 1 follow-up comments (AMT-2022-174)

Thank you for your comments. Our responses are below.

Minor Comments
Comment 1. L88: Can you cite the microphysics processing software used (DOI or previous paper)? The microphysics data readme for that dataset (L432) doesn't seem to specify this.
Response: The ICICLE data was processed using the Environment and Climate Change Canada (ECCC) D2G software. Currently, no formal reference for this software is available.

Comment 2. L93–L94: Are these the bin midpoints? As it reads, there is a gap in PSD sizes. Specifying the bin left (right) endpoint for the lowest (highest) size used for each probe might avoid confusion.

Response: These are midpoints of size bins (not left endpoints). Numerical simulations performed by the second author (not published, in preparation for publication) showed that for OAPs the midpoints of the size bins, starting from the third bin, with a high accuracy can be presented as $D_{mid}=kd$, where k is the size bin number, and d is the width of the size bin. There appears to be no gaps in combined PSDs.

Comment 3. L189: This may be true for the DWR pair used here based on the ICICLE observations, but other studies (Kneifel et al. 2015; Chase et al. 2018; Mason et al. 2019), albeit triple-frequency studies, do suggest sensitivity in particular DWR pairings as the result of riming or effective density. Maybe specify that your statement is valid for the radar frequencies and types of cloud sampled in this study?

Response: We specified the cloud conditions considered in this study. We specifically mentioned that we sampled volumes with only traces of supercooled LWC. The influence of effective density on individual DWR is not expected to be very significant given uncertainties of measurements.

Technical Corrections

Comment 1. L82–L84: "(CDP)(Lance et al. 2010)" -> "(CDP, Lance et al. 2010)" and elsewhere to be consistent with nomenclature on L96.
Responses: corrected as suggested.

Comment: 2. L108: There should be a space between "mm^6" and "m^-3".
Response: the space was added.

Comment: 3. L234: re should be a space between "g" and "m^-3".
Response: the space was added.

Comment: 4. L345: Do you mean to say "numerator"?
Response: Yes, "numerator".

Responses to the reviewer 2 follow-up comments (AMT-2022-174)

Thank you for your comments. Our responses are bellow.

Minor comments:

Comment1: l. 152-156: Regarding the observation that (because of non-Rayleigh scattering) reflectivity is lower in the horizontal than in the vertical beam measurements: is this also related to the fact that horizontal polarization is used? If a vertical polarization were used, would this also be expected?
Response: Yes, and the magnitude of the zenith enhancement increase is expected to be even higher for vertically polarized signals (because vertical dimensions of particles are generally smaller than the horizontal ones). In the earlier study (Matrosov et al. 2012), the slant linear 45 deg polarization was used, rather than the horizontal polarization. The corresponding zenith reflectivity enhancements were present and for highly non-spherical particles they were greater than 10 dB.
Comment 2.- Fig. 7: Any specific reason why this plot is semi-log, compared to the previous ones? It makes it more difficult to compare directly the quality of the relations (especially for small particles, since there are very few Dv values below 1mm).
Response: For the most part it is because the characteristic size – reflectivity relations were first published in the log scale (e.g., Matrosov and Heymsfield 2017, DOI: 10.1175/JAMC-D-170076.1). The use of similar scales could facilitate comparisons of relations from this study and the earlier ones obtained for different wavelengths. We added this earlier paper to the reference list.

Technical corrections:
Comment: l. 15 "predominantly"
Response: corrected.
Comment: l. 19 "W-band band"
Response: corrected.
Comment: l. 25->28 the sentence feels a bit clumsy. Consider rewording it.
Response: This sentence was modified.
Comment: l. 112 "the influence"
Response: corrected as suggested.
Comment: l. 164 "irregularly shaped"
Response: corrected as suggested.
Comment: l. 167 "The habit recognition was tuned so that".. or "in a way that.."
Response: corrected as suggested.
Comment: l. 170 "vertically" and "horizontally"
Response: corrected as suggested.
Comment: l. 176 "The amplitude/ magnitude of these differences"?
Response: The W-band bright band enhancement in this example was ~ 3dB while at X-band it was ~10 dB or so. The corresponding information was addedi n the paper.
Comment: l. 206 reference to fig 3 is to give an example where we see dendrites corresponding to a high DWR values, and not a general statement. To avoid confusion, I would include "see for example the images in Fig. 3" (or alike)
Response: It is now specified in the paper that this statement refers to this particular example.
Comment: l. 341 "i.e., it is proportional.."
Response: corrected as suggested.

---

## Author Response (AR3)

Responses to the editor comments.
Alexis, Thank you for your feedback. Below are the responses.

Comment: - Mention the soft used to process the in situ data (even if not published)
Response: It was added in the text that "The microphysical data were processed using the ECCC D2G software". Note that the D2G is not an abbreviation. Because of that it is not spelled out.

Comment: - Mention around l. 93-94 that the values provided are midpoints of the size classes (to avoid the misinterpretation of discontinuous PSD)
Response: the corresponding information was added in text.

Comment: - Add around l.189 that the DWR is not much influenced by density in the conditions under study here. I agree with Reviewer1 that if not mentioned, this could appear as a general statement, as the conditions of this study are only specified later on (l.116-117).
Response: The manuscript already states on lines 189-190 that "One of the features of DWR, is that this ratio exhibits only a relatively weak dependence on particle bulk density (Matrosov et al. 2019)". This feature is valid not only for the conditions of this study but it is more general.